# COVID-19 and Integrated Multidisciplinary Care Model in Parkinson’s Disease: Literature Review & Future Perspectives

**DOI:** 10.3390/bs12110447

**Published:** 2022-11-13

**Authors:** Seyed-Mohammad Fereshtehnejad, Mayela Rodríguez-Violante, Monica S. Ponce-Rivera, Daniel Martinez-Ramirez, Adolfo Ramirez-Zamora

**Affiliations:** 1Division of Neurology, Department of Medicine, University of Ottawa, Ottawa, ON K1N 6N5, Canada; 2Division of Clinical Geriatrics, Department of Neurobiology, Care Sciences and Society (NVS), Karolinska Institutet, 14186 Stockholm, Sweden; 3Movement Disorders Clinic, Clinical Neurodegenerative Research Unit, Mexico City 05300, Mexico; 4Tecnologico de Monterrey, Escuela de Medicina y Ciencias de la Salud, Monterrey 66220, Mexico; 5Norman Fixel Institute for Neurological Diseases, University of Florida, Gainesville, FL 32611, USA

**Keywords:** Parkinson’s disease, multidisciplinary care model, COVID-19

## Abstract

Clinical diversity and multi-systemic manifestations of Parkinson’s disease (PD) necessitate the involvement of several healthcare professionals from different disciplines for optimal care. Clinical guidelines recommend that all persons with PD should have access to a broad range of medical and allied health professionals to implement an efficient and effective multidisciplinary care model. This is well supported by growing evidence showing the benefits of multidisciplinary interventions on improving quality of life and disease progression in PD. However, a “multidisciplinary” approach requires gathering healthcare professionals from different disciplines into an integrative platform for collaborative teamwork. With the Coronavirus Disease 2019 (COVID-19) pandemic, implementation of such a multidisciplinary care model has become increasingly challenging due to social distancing mandates, isolation and quarantine, clinics cancellation, among others. To address this problem, multidisciplinary teams are developing innovate virtual platforms to maintain care of people with PD. In the present review, we cover aspects on how SARS-CoV-2 has affected people with PD, their caregivers, and care team members. We also review current evidence on the importance of maintaining patient-centered care in the era of social distancing, and how can we utilize telehealth and innovative virtual platforms for multidisciplinary care in PD.

## 1. Introduction

Daily life activities have changed after the coronavirus disease 2019 (COVID-19) pandemic caused by the SARS-CoV-2 virus [1]. People had to adjust to a virtual environment in daily life, in areas including schooling, employment, and extracurricular activities. This transition has not been easy for people with Parkinson’s disease (PwP) and their caregivers [2]. Parkinson’s disease (PD) is a complex neurological disorder with a variety of motor and non-motor symptoms that requires close monitoring by treating physicians complemented with physical activities and rehabilitation therapies to improve patient quality of life [3].

One of the main recommendations to prevent the spread of this new virus is social distancing measures. Social distancing and lockdown restrictions had negative impacts on PwP [2,4]. First, clinical and rehabilitation therapy appointments needed to be rescheduled or cancelled. Second, PwP who are isolated from family and friends experience an increase in psychological distress. Although many clinicians have implemented telemedicine as a tool to continue providing care, this approach is not available to many patients, including underserved minorities or patients with limited technological literacy [5]. To address this problem, multidisciplinary teams are innovating virtual platforms to maintain care for PwP. In the present review, we will cover aspects on how COVID-19 has impacted PwP, caregivers, and care team members. We will also discuss the importance of maintaining patient-centered care in the era of social distancing, how telehealth and remote care can be utilized in the management of PwP, and innovative virtual platforms for multidisciplinary care in PD.

### 1.1. Impact of COVID-19 on People with Parkinson’s Disease

The COVID-19 pandemic has affected PwP in various ways (Figure 1). 

PwP reported worsening motor symptoms, but more importantly, their non-motor symptoms such as depression, anxiety, and/or apathy have also been affected during the pandemic [6]. Some of the postulated reasons are social distancing measures, cancellation of in-person follow-up visits and rehabilitation therapies, limited activities, stigma and social stressors. Figure 1 illustrates the direct and indirect impact of the COVID-19 pandemic on various aspects of care in PD. PwP have been undertaking less physical activity than before the lockdown. A study performed in Italy during the first wave of the pandemic showed that up to 60% of PwP experienced a significant worsening of their general conditions, which mainly resulted from the reduction of physical activity due to the lockdown. However, nearly half of the PwP in this study adapted new habits or started using technology assistance tools to continue practicing physical activity [7]. Self-reported questionnaires suggest clinical worsening by increasing non-motor and motor scores in daily life activities using the Movement Disorder Society-Unified PD Rating Scale (MDS-UPDRS) in PwP during social distancing, particularly in areas with limited access to telemedicine. A self-reported study in Italy showed that seeking information about clinical services, acute clinical worsening, or the relationship between PD and COVID-19 were the most common needs or queries raised by PwP during the first two weeks of the COVID-19 lockdown in 2020 [8]. This study highlights the need for a rapid response system via telecommunication to address these concerns.

Another care-related aspect that has been adversely affected in PwP during the pandemic is the access to PD medications. In one international survey, 45.4% of PwP reported that their access to PD medications had been affected during the COVID-19 pandemic, particularly in resource-poor countries [9]. An online web-based survey also demonstrated a more disrupted access to medications during the pandemic in PwP who reside in lower income countries together with loneliness and non-Whites [10]. As a consequence, disrupted access to PD medications results in the deterioration of symptomatic control, which contributes to worsening of motor and non-motor features in PwP [9]. A similar adverse effect has also been reported for advanced therapies in PwP. A telephone-based survey showed worsening of dystonia in PwP treated with deep brain stimulation (DBS) due to problems with DBS management and programming during the lockdown restrictions [11].

People with chronic conditions including PD rely on community and social networks to maintain their health and well-being. Reports using large surveys revealed worsening in functioning (24.8–37.3%), health (33.8–43%), and well-being (26.1–47.1%) in PD during the COVID-19 pandemic [12]. Additionally, increased emotional stress, media coverage of the pandemic, and uncertainty of the future can further worsen clinical features in PwP [13,14]. In another survey aiming to better understand the emotional and behavioral consequences of the public health policies implemented to mitigate COVID-19, 50% of respondents reported a negative change in PD symptoms, with 45–66% reporting mood disturbances [2]. Social distancing has also affected families and caregivers, causing them additional stress and fatigue, isolation and burn out [15], which will be discussed in the next section.

PwP have the same probability of becoming infected with the disease when compared to people without PD [16]. However, when affected by SARS-CoV-2, PwP show worsening of motor and non-motor symptoms with increased daily off-time. Whether or not this worsening of PD symptoms is related to acute systemic inflammation and/or brain involvement is not quite clear. It has been speculated that the SARS-CoV2 virus may directly or indirectly induce immune–inflammatory events which affect the central nervous system (CNS) [17]. Findings from autopsy studies suggest that SARS-CoV-2 can penetrate the olfactory mucosa and may enter the brain along the olfactory tract [18]. Specimens from post-mortem studies have shown the direct invasion of brain tissue by SARS-CoV-2 virus in various regions, including the olfactory bulb, amygdala, entorhinal area, temporal and frontal neocortex, dorsal medulla and leptomeninges [19]. Current evidence, however, is not quite conclusive with regard to whether COVID-19 involvement of CNS can directly affect PD manifestation.

Patients with advanced PD have shown poorer outcomes including severe illness, pneumonia, need for supplemental oxygen, hospitalization, and a higher risk of mortality. The latter is associated with older age and longer disease duration [10,20]. A recent study of an administrative claims database covering a large number of hospitals in Germany in 2020 showed that the COVID-19 inpatient mortality rate was significantly higher in hospitalized PwP (35.4%) than in non-PD patients admitted to the hospital (20.7%) [21]. The increased mortality rate in hospitalized PwP was especially prominent in patients aged 75–79 years. In addition, overall inpatient mortality of PwP was significantly higher in 2020 than in 2019 (5.7% vs. 4.9%), which could be attributed to the pandemic [21].

COVID-19 stressors such as loss of social contact, watching COVID-19 media, and limitations on exercise or outdoor activities had increased psychological distress, which aggravates PD severity. One of the most burdensome stressors reported is not being able to attend the funeral or visiting the hospital of a loved one. Personality traits and social support play a crucial role in how much the stressor exposure load will affect PwP. Most vulnerable PD patients have high levels of anxiety, rumination, and neuroticism and lower scores on social support, optimism, and resilience [20].

### 1.2. Impact of COVID-19 on Caregivers

The COVID-19 pandemic has resulted in unprecedented disruptions in various aspects of the integrated multidisciplinary care of PwP. The pandemic is still dynamically evolving at the time of writing this review, and as such, it should be considered that research studies covering different pandemic eras will inevitably reflect different effects on patients and caregivers. From the beginning of the pandemic, with prioritizing acute care in the forefront of the healthcare sector, a majority of the in-person multidisciplinary clinical care platforms have been put on hold. As such, an extra care burden is now on the shoulders of the caregivers of PwP during the COVID-19 era. In one survey conducted in India following three weeks of lockdown during the first COVID-19 endemic wave, caregivers of PwP were found to be well informed and coping well with the effects of the pandemic on PwP, yet this was an early reflection at the beginning of the pandemic [22]. With a longer lockdown and cancellation of regular in-person visits at multidisciplinary PD clinics, caregivers reported a significantly increased burden due to worsening motor symptoms of PwP resulting from the inability to access healthcare and difficulty obtaining medications [23]. In another study, nearly 60% of the caregivers of PwP expressed concerns about anxiety, with 4.2% demonstrating severe anxiety during the COVID-19 pandemic [24]. Using structured telephone interviews during the last 10 days of the first-wave lockdown in Italy, researchers showed that the level of stress worsened in 43.8% of patients with PD and in 53.1% of caregivers [25]. The higher level of stress and extra burden among caregivers during the lockdown period were mainly attributed to the worsening of non-motor features (particularly neuropsychiatric, autonomic and cognitive domains) in PwP, demonstrated by the significant increase in Non-Motor Symptoms Scale (NMSS) scores [25]. Under strict home confinement during the COVID-19 pandemic, the worsening of motor and non-motor symptoms of PD significantly affect not only the patients’ quality of life but also their caregivers’ burden.

### 1.3. Telehealth and Remote Management

For the previous two decades before the COVID-19 pandemic, there was already a surge in the concept of telemedicine in general, and in neurology (teleneurology), which has shown replicable outcomes compared to traditional face-to-face consultation [26]. The use of telemedicine and telehealth have risen exponentially during the pandemic.

Telehealth can be broadly defined as the provision of health care remotely by means of information and communication technologies [27]. These technologies include, among others, tools such as mobile devices, email, short message service (SMS), and dedicated online platforms. Telemedicine is centered in medical care alone, while telehealth encompasses other uses, such as tele-education and tele-rehabilitation. The term tele-neurology has been coined to refer exclusively to the use of telehealth in persons with a neurological disorder [28].

The feasibility of telemedicine for the medical care of PwP has been demonstrated by Dorsey et al., who reported that between 93 and 97% of telemedicine consultations are successfully completed [29,30]. Tarolli et al. have recently reported that remote visits are reliable in the context of phase III clinical trials involving individuals with early PD [31]; and Dorsey et al. favored the use of virtual research visits for self-reported diagnosis of PD [32]. Tanner et al. reported a 96% retention rate using mailed self-administered questionnaires and telephone calls for the long-term follow-up of PD patients in a clinical study [33]. Feasibility is a requirement, but patient satisfaction and, more importantly, clinical benefit has been less studied. Hanson et al. reported that patients with a movement disorder diagnosis were highly satisfied with the use of telemedicine, even favoring future telemedicine visits when feasible [34]. Seritan et al. reported that 95% of a small sample of PwP were highly satisfied after a telepsychiatry visit [35]. For a virtual clinical assessment, web-based versions of the Unified Parkinson’s Disease Rating Scale (UPDRS), the Unified Dyskinesia Rating scale (UDRS), and the Non-Motor Symptoms Questionnaire (NMSQuest) have been deemed to be feasible [36]. Schneider et al. assessed the feasibility and validity of conducting a modified online version of the MDS-UPDRS motor, without assessment of rigidity or postural stability during remote visits, and reported a 98% success rate [37]. A recent meta-analysis of telehealth interventions in PD showed benefits in the improvement of motor impairment [38].

Telemonitoring is also an important part of telehealth, especially for PwP. Even predating the COVID pandemic, various methods including body-worn sensors, home sensors, smartphone apps, digital diaries, or the analysis of common appliances such as computer keyboards have been used for the telemonitoring of symptoms in PD [39]. The remote assessment of speech features has been tested for the diagnosis of PD, with preliminary studies showing an accuracy above 90% [40,41]. Wearable sensor devices have also gained interest in this field, and portable accelerometers and gyroscopes have been tested to quantify motor symptoms, but further validation is still needed to develop adequate machine learning algorithms [42]. Telehealth in PD has proved to be cost-effective [43]. However, the effectiveness of many of these telemonitoring and teletreatment programs have not been fully investigated, which necessitates further studies in the future [39].

### 1.4. Patient-Centered and Integrated Multidisciplinary Care in the Era of Social Distancing

Clinical guidelines recommend that all persons with PD should have access to a broad range of medical and allied health professionals for personalized, integrative, and multidisciplinary care [44]. The availability of a health professional and alternative therapies has been limited during the COVID-19 pandemic, with increasing difficulties accessing health providers needed in PD care. Several causes of disruption have been identified: closure of inpatient or outpatient services or consultations as per health authority directive, decrease in outpatient volume due to patients not presenting for care, a decreased volume of patients due to cancellation of elective care, limited inpatient services due to saturation, insufficient staff to provide services, clinical staff shifted to provide COVID-19 clinical management or emergency support, insufficient Personal Protective Equipment (PPE) available for health care professionals to provide services, the unavailability or stock of essential medicines, medical diagnostics or other health products at health facilities, and travel restrictions hindering access to the health facilities for patients [45]. These concerns are particularly relevant in rural areas and in regions with limited access to telemedicine.

A potential alternative and management approach relies on the implementation of virtual, telehealth visits. The rapid adoption of telehealth services to deliver clinical care is supported by both the healthcare practitioners and the PD community, and demonstrates the feasibility of the use and access of these services [46]. This approach could expand beyond the management of medications and acute changes in symptoms and include discussions on lifestyle/wellness, support groups, diet, rehabilitation, exercise, and access to mental health providers. Patients are usually eager to engage with instructors and allied providers. Virtual support groups are using video-conference technology serves as model interventions to keep PwP connected, educated, and empowered. A large number of participants can easily access experts and exchange ideas and concerns with people in the group. Sessions can be recorded for other patients to rewatch at their convenience. Almost half (46%) of PwP preferred to continue using telehealth after the coronavirus outbreak has ended [2]. There has been a consistent and necessary increase in telemedicine, with some centers reporting an increase from 9.7% use prior to the pandemic to 63.5% during the pandemic [2]. Despite the benefits of telemedicine, telehealth poses challenges for reaching less technology proficient aging populations and underrepresented minority populations [46]. A recent Parkinson’s Foundation survey indicates that these challenges may be further augmented by low socioeconomic or educational status [2]. Importantly, the availability of specific treatments that can reduce stress, such as mindfulness-based intervention, is critical during the pandemic. Several studies have shown that mindfulness can reduce depression and anxiety, and even improve motor symptoms in PwP [47].

Telemedicine and the remote administration of rehabilitative treatment (telerehabilitation) has emerged as an excellent alternative to bypass physical distancing barriers in neurorehabilitation. Advantages of telerehabilitation include its ability to circumvent physical barriers, transportation issues (decrease travel expenses and caregiver burden) and financial concerns. A loss of aerobic exercise during the COVID-19 pandemic may lead to worsening of motor and non-motor manifestations such as insomnia or constipation. Reduced physical exercise may contribute to increased psychological stress. Previous studies in stroke suggest that telerehabilitation may have a positive impact on a range of primary and secondary neurological outcomes despite the large variability in administration and implementation of protocols [48]. The effectiveness of telerehabilitation in PD can be increased by new technologies such as virtual reality, wearable sensors and signal processing using machine learning to predict the long-term dynamics of patient recovery [49]. Initial response to the transition to telerehabilitation has been halted by lack of reimbursements, but early studies suggest its feasibility and encouraging results [49]. Further research to determine how it compares with in person visits and to determine best candidates are needed.

Telemedicine can also provide access to additional services such as speech therapies during the COVID-19 pandemic. The validation of online assessment of speech disorders in PD has been reported using videoconferencing [50]. Interventional studies have focused on the delivery of the intensive Lee Silverman Voice Treatment (LSVT) program and demonstrated the noninferiority of this treatment when delivered remotely [51]. Challenges to this approach steam from reimbursement for services, adequate aided hearing and vision during visits, and the availability of assistance by the caregiver when needed [52].

A particular challenge related to social distancing in the pandemic relates to the management of advanced therapies in PD including levodopa intestinal gel infusion (Duopa) or DBS. Patients might benefit from tertiary centers’ expertise and the rapid implementation of telemedicine visits. Product manufacturer specialists can provide acute care and support either in person or virtually. Providers can also rely on caregivers and patients to perform adjustments on their devices at home with physician guidance using the unlock features of the device or adding stimulation ranges to DBS programming groups. Patients should be educated on how to use their own patient controllers to allow for adjustments and to perform battery checks. A unique challenge might be the need to manage patients living in nursing homes when local quarantine restrictions might limit interactions with other providers and nursing staff [53]. Recently, during the COVID-pandemic, DBS tele-programming adjustment sessions were implemented in China with satisfactory results. The main DBS-related adjustment was voltage change to optimize symptom control, which was performed satisfactory with no significant side effect via telemedicine [54]. In another study on post-surgical DBS patients, remote programming based on the online evaluation of patient symptoms showed promising results with near complete satisfaction [55].

Despite all of the above advantages, there are limitations on the use of telemedicine. The absence of in person contact can potentially impact the verification of the safety of procedures and damage physician-patient relationships and trust. Additionally, there are concerns about the largely heterogenous outcomes of an intervention studied with different assessments and benefits in diverse populations. This will certainly limit global applicability [49]. Some populations including underserved (low income) populations might encounter significant challenges using telehealth, as the rates of low technological literacy [2], limited access to internet connection, and the lack of technical assistance in the case of malfunctioning are higher.

### 1.5. Innovative Platforms for Multidisciplinary Care

In PD, telenursery [56], telehealth neuropsychology [57], and telerehabilitation [58] have been used along with teleneurology and telepsychiatry as components of a multidisciplinary care model. Consequently, multidisciplinary care in the setting of virtual visits should not differ from the traditional in-office visits. Multidisciplinary tele-counseling involving neurological, neuropsychological and nutritional consultations have been shown to be feasible for elderly care [59].

Electronic health records are now commonly used, but an expansion of their capabilities is required, including embedding home monitoring systems and patient reported outcomes [60]. Assembling a virtual multidisciplinary team requires health-related professionals willing to adapt to this informal virtual setting. It has been reported that clinician personality and interpersonal preferences are correlated with the appeal, satisfaction, and perceived effectiveness of telemedicine [61]. Members of the multidisciplinary team must be familiar with telehealth tools but also be convinced of their advantages and aware of the limitations.

New integrated multidisciplinary care models have been emerging to improve quality of care for PwP in the era of the COVID pandemic with its new challenges. An example is a model called the ’physician-advanced practice provider (APP) team model’, recently developed for subspecialty care such as movement disorders in an academic center in the USA [62]. This model was developed in response to the long waiting times caused by the shortage of neurologists and the growing number of patients waiting to be seen in a movement disorders clinic. As seen in many countries, this situation has been made even worse during the pandemic. The APP team model consists of a clinic staff, a social worker, trained practice providers, and residents. History taking, physical examination and documentation are performed by the advanced practice providers and residents, while the movement disorders specialist oversees issues related to diagnosis and designs a multidisciplinary individualized management plan for each patient. This model was shown to significantly improve access to subspecialty care for PwP with high patient and team member satisfaction [62]. A recent review on the efficiency of virtual care for patients with Alzheimer’s disease during the COVID pandemic concluded that successful telemedicine programs are those with the involvement of interdisciplinary teams to manage patient complexity, medical and psychiatric comorbidities, and psychosocial needs with a progressive neurodegenerative condition like Alzheimer’s dementia or Parkinson’s disease [63]. 

The use of mobile devices, apps or wearable sensors would be desirable. Yet, access to technology and high-speed internet connection to assure good audio and video quality to carry out a remote neurological exam is the most important limitation of telemedicine, especially in developing countries. Finally, technology and electronic literacy and regulatory issues should be considered before implementing a virtual platform. For individuals with neurodegenerative conditions such as PD, there are additional challenges using the virtual platforms of telemedicine. Some examples include physical obstacles in handling electronic devices (e.g., hand tremor, dexterity problem), lack of familiarity with how these tools are used in this predominantly elderly population, and reduced executive functioning in advanced stages of PD with cognitive impairment, which affects the feasibility of using computer equipment. Lastly, the patient-doctor relationship and shared decision making rely on a high level of communication, thus a hybrid model of virtual and in-person visits would offer the best of both worlds.

## 2. Conclusions

Besides its direct effects on motor and non-motor features of PD and caregivers’ burden, the COVID-19 pandemic is also causing world-wide social dislocation resulting in operational challenges for an ideal multidisciplinary care model of PwP. In response, telehealth has rapidly moved to the frontline of clinical practice (including neurology) during the COVID-19 pandemic, enabling PwP to access healthcare services remotely. Although innovative technology is emerging to ease the implementation of a virtual integrative multidisciplinary care for PwP, further attempts are required to expand the accessibility of these platforms in poor resource regions worldwide.

## Figures and Tables

**Figure 1 behavsci-12-00447-f001:**
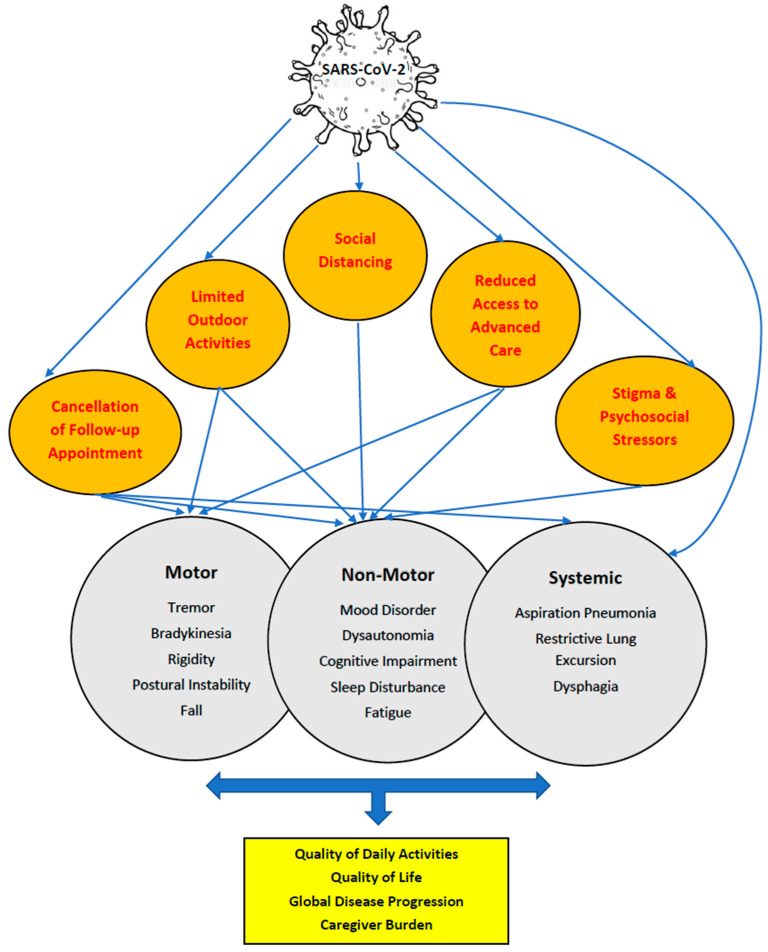
Various aspects of COVID-19 pandemic affecting different aspects of care in Parkinson’s disease (PD) with direct or indirect impact on PD manifestations (motor and non-motor), systemic health and eventually quality of life and their caregiver burden.

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
