# Peer review of "COVID-19 and Integrated Multidisciplinary Care Model in Parkinson’s Disease: Literature Review & Future Perspectives"

_behavsci, 2022, doi:10.3390/bs12110447_

Round 1
Reviewer 1 Report
Grammar and style are deficient. A simple automated grammar check would show this. For example, the following 1.5 sentences contain multiple errors: Second, PwP isolated from family and friends increasing psychological distress. Although most of clinicians The sentence is not a sentence and "most of clinicians" is not English. The tenses change randomly, etc. Next, it is inadvisable to make claims such as attributing worsening of symptoms to "acute systemic inflammation or changes in drug pharmacokinetics." This is just speculation, and the further speculation that CoV-2 invasion of the CNS "might induce...PD, in the future" is unsupported and irrelevant to the topic of the paper. More attention should be paid to the special challenges of telehealth for PD patients: physical challenges in handling computer equipment; lack of familiarity with how these tools are used in this predominantly elderly population; and dementia in a substantial subpopulation. Some of this is discussed in the last paragraph before the conclusion but it should receive much more consideration. Systems that can help with some of these challenges exist in China and are not discussed here (e.g., remote video and DBS programming from SceneRay and PINS, which might be adapted to non-DBS uses as well).
Author Response
Reviewer #1
Grammar and style are deficient. A simple automated grammar check would show this. For example, the following 1.5 sentences contain multiple errors: Second, PwP isolated from family and friends increasing psychological distress. Although most of clinicians The sentence is not a sentence and "most of clinicians" is not English. The tenses change randomly, etc.
Thank you for raising this issue. As per the comment, we have now edited the grammar and language style of the manuscript. All revisions are highlighted with tracked changes in the revised version. We specifically corrected the above-mentioned statements in this comment.
Next, it is inadvisable to make claims such as attributing worsening of symptoms to "acute systemic inflammation or changes in drug pharmacokinetics." This is just speculation, and the further speculation that CoV-2 invasion of the CNS "might induce...PD, in the future" is unsupported and irrelevant to the topic of the paper.
We truly appreciate this very relevant comment. As per suggestion, we have completely revised this paragraph in our revised manuscript. We deleted some of these statements which are not completely supported by current evidence (such as “might induce...PD, in the future”). We clarified other speculations as not proven yet. Also, we added new evidence from postmortem studies on the effect of COVID-19 on CNS. The revised paragraph is as follows:
“Whether or not this worsening of PD symptoms is related to acute systemic inflammation and/or brain involvement is not quite clear. It has been speculated that SARS-CoV2 virus may directly or indirectly induce immune–inflammatory events which affect central nervous system (CNS) [17]. Findings from autopsy studies suggest that SARS-CoV-2 can penetrate the olfactory mucosa and may enter the brain along the olfactory tract [18]. Specimens from some post-mortem studies have shown direct invasion of brain tissue by SARS-CoV-2 virus in various regions namely olfactory bulb, amygdala, entorhinal area, temporal and frontal neocortex, dorsal medulla and leptomeninges [19]. Current evidence, however, is not quite conclusive whether COVID-19 involvement of CNS can directly affect PD manifestations.”
More attention should be paid to the special challenges of telehealth for PD patients: physical challenges in handling computer equipment; lack of familiarity with how these tools are used in this predominantly elderly population; and dementia in a substantial subpopulation. Some of this is discussed in the last paragraph before the conclusion but it should receive much more consideration.
We highly appreciate this relevant comment. As per suggestion, we have now added a new paragraph to the last section before ‘conclusion’, to highlight these obstacles and challenges as follows:
“For individuals with neurodegenerative conditions such as PD, there are additional challenges using virtual platforms of telemedicine. Some examples include physical obstacles in handling electronic devices (e.g. hand tremor, dexterity problem), lack of familiarity with how these tools are used in this predominantly elderly population, and reduced executive functioning in advanced stages of PD with cognitive impairment which affect feasibility of using computer equipment.”
Systems that can help with some of these challenges exist in China and are not discussed here (e.g., remote video and DBS programming from SceneRay and PINS, which might be adapted to non-DBS uses as well).
This is a very good suggestion as well. As recommended, we have now added more sentences to summarize results from studies on remote programming of DBS tools during the pandemic as follows:
“Recently during the COVID-pandemic, DBS tele-programming adjustment sessions were implemented in China with satisfactory results. The main DBS-related adjustment was voltage change to optimize symptom control which was performed satisfactory with no significant side effect via telemedicine [54]. In another study on post-surgical DBS patients, remote programming based on online evaluation of patient’s symptoms showed promising results with near complete satisfaction [55].”

Reviewer 2 Report
Fereshtehnejad et al. performed a narrative review investigating the effect of the COVID-19 pandemic on the health care of people with Parkinson Disease (PwP) and their caregivers and proposed a new “Patient-centered and Integrated Multidisciplinary Care” for the future management of PwP during such a period. The review reads well, but the topic is not original. I propose some suggestions for the improvement of the manuscript.
- The authors should insert some Figures/Tables in order to give a graphical summary of the evidence emerging from this review.
- The authors should focus more on the impact of the COVID-19 pandemic on the management of PwP and their caregivers, such as the reduced access of PwP to PD’s medications (PMID: 32860226), difficulty in managing second-line therapies (PMID: 33391174), differences between high- and low-income patients (PMID: 32925107), and many others.
- “PwP have the same probability of becoming infected with the disease when compared to people without PD. On the other hand, PwP infected by SARS-CoV-2 show a worsening of motor and non-motor symptoms and increased in their daily off-time, prob-ably due to acute systemic inflammation or changes in drug pharmacokinetics.” Please add some references.
- I think that the section about the future perspective is too long and does not add any critical information on current knowledge on this subject.
In conclusion, the review is well written, but the summary of the previous evidence on this topic is cursory and does not provide any additional information to that already available in the current literature. I believe that an integration of this point and a graphical representation of the results emerging from this review are mandatory for the publication of this article.
Author Response
Reviewer #2
Fereshtehnejad et al. performed a narrative review investigating the effect of the COVID-19 pandemic on the health care of people with Parkinson Disease (PwP) and their caregivers and proposed a new “Patient-centered and Integrated Multidisciplinary Care” for the future management of PwP during such a period. The review reads well, but the topic is not original. I propose some suggestions for the improvement of the manuscript.
The authors should insert some Figures/Tables in order to give a graphical summary of the evidence emerging from this review.
This is indeed an excellent suggestion. As per recommendation, we have now created a new Figure to schematically summarize the impact of COVID-19 pandemic on various aspects of Parkinson’s disease, to illustrate the way it directly or indirectly impacts their quality of life and caregiver burden. Please refer to Figure 1 in the revised version of our article.
The authors should focus more on the impact of the COVID-19 pandemic on the management of PwP and their caregivers, such as the reduced access of PwP to PD’s medications (PMID: 32860226), difficulty in managing second-line therapies (PMID: 33391174), differences between high- and low-income patients (PMID: 32925107), and many others.
We appreciate this very relevant comment. As per suggestion, we have now added a new paragraph to summarize the above-mentioned studies on the impact of pandemic on PD medication access. All these new citations are added to the list of references in the revised manuscript. The paragraph is as follows:
“Another care-related aspect that has been adversely affected in PwP during pandemic is the access to PD medications. In one international survey, 45.4% of PwP reported that their access to PD medications had been affected during COVID-19 pandemic, particularly in resource-poor countries [10]. An online web-based survey also demonstrated a more disrupted access to medications during pandemic in PwP who reside in lower income countries together with loneliness and non-White race [7]. As a consequence, a disrupted access to PD medications results in deterioration of symptomatic control, which contributes to worsening of motor and non-motor features in PwP [10]. A similar adverse impact has also been reported for advanced therapies in PwP. A telephone-based survey showed worsening of dystonia in PwP treated with deep brain stimulation (DBS), due to problems with DBS management and programming during the lockdown restriction [11].”
“PwP have the same probability of becoming infected with the disease when compared to people without PD. On the other hand, PwP infected by SARS-CoV-2 show a worsening of motor and non-motor symptoms and increased in their daily off-time, prob-ably due to acute systemic inflammation or changes in drug pharmacokinetics.” Please add some references.
As per the suggestion, we added a new citation for the first part of this paragraph: “PwP have the same probability of becoming infected with the disease when compared to people without PD [16]”. The citation is: Artusi CA, et al. COVID-19 and Parkinson's Disease: What Do We Know So Far? J Parkinsons Dis. 2021;11(2):445-454.
As recommended by the first reviewer as well, the second part of this paragraph is not quite supported by current evidence. As such, we have completely revised this statement and added new evidence from other references for this speculation of CNS involvement by COVID-19 infection. The revised paragraph is as follows:
“PwP have the same probability of becoming infected with the disease when compared to people without PD [16]. However, when affected by SARS-CoV-2, PwP show worsening of motor and non-motor symptoms with increased daily off-time. Whether or not this worsening of PD symptoms is related to acute systemic inflammation and/or brain involvement is not quite clear. It has been speculated that SARS-CoV2 virus may directly or indirectly induce immune–inflammatory events which affect central nervous system (CNS) [17]. Findings from autopsy studies suggest that SARS-CoV-2 can penetrate the olfactory mucosa and may enter the brain along the olfactory tract [18]. Specimens from some post-mortem studies have shown direct invasion of brain tissue by SARS-CoV-2 virus in various regions namely olfactory bulb, amygdala, entorhinal area, temporal and frontal neocortex, dorsal medulla and leptomeninges [19]. Current evidence, however, is not quite conclusive whether COVID-19 involvement of CNS can directly affect PD manifestations.”
I think that the section about the future perspective is too long and does not add any critical information on current knowledge on this subject.
We appreciate reviewer’s suggestion. As recommended, we have now shortened the section on ‘Innovative Platforms for Multidisciplinary Care’ with deletion of redundant statements.
In conclusion, the review is well written, but the summary of the previous evidence on this topic is cursory and does not provide any additional information to that already available in the current literature. I believe that an integration of this point and a graphical representation of the results emerging from this review are mandatory for the publication of this article.
We appreciate reviewer’s comments. As mentioned above, we have now revised our article taking into account all comments and suggestions. In particular, a new Figure is added to illustrate how COVID pandemic impact different aspects of Parkinson’s disease in PwP. In addition, we have added new paragraphs to summarize results from further relevant studies with more details on their findings.

Round 2
Reviewer 1 Report
The authors have undertaken a substantial rewriting of this report. It is much improved, much more informative and concise, and I believe it is worthy of publication in this form.
Reviewer 2 Report
The manuscript has been significantly improved in this revised version.